# Risk of Acute Anterior Uveitis in Ankylosing Spondylitis According to the Type of Tumor Necrosis Factor-Alpha Inhibitor and History of Uveitis: A Nationwide Population-Based Study

**DOI:** 10.3390/jcm11030631

**Published:** 2022-01-26

**Authors:** Soo Min Ahn, Minju Kim, Ye-Jee Kim, Yusun Lee, Yong-Gil Kim

**Affiliations:** 1Department of Rheumatology, Asan Medical Center, University of Ulsan College of Medicine, Seoul 05505, Korea; lisianthus@amc.seoul.kr; 2Department of Clinical Epidemiology and Biostatistics, Asan Medical Center, University of Ulsan College of Medicine, Seoul 05505, Korea; minjukim@amc.seoul.kr (M.K.); kimyejee@amc.seoul.kr (Y.-J.K.); 3AbbVie Pty, Ltd., Seoul 06182, Korea; radiotherapysunny@gmail.com

**Keywords:** ankylosing spondylitis, uveitis, tumor necrosis factor inhibitors, adalimumab, infliximab, golimumab, etanercept

## Abstract

Background: We evaluated the risk of acute anterior uveitis (AAU) in patients with ankylosing spondylitis (AS) during treatment with tumor necrosis factor-alpha inhibitors (TNFis). Methods: This study was performed on AS patients using the Korean National Health Insurance claims database. We analyzed the first and total occurrence of AAU during the first 2 years of TNFis use according to the type of TNFis. Additionally, the occurrence of AAU was assessed in subgroups with or without prior AAU before TNFis initiation. Results: In total, 5938 AS patients initiated TNFis use between 2009 and 2017 and used them for more than 2 years. Among them, 1488 (25.1%) patients had a history of AAU before starting TNFis treatment. Compared to adalimumab, the use of etanercept (hazard ratio [HR] 1.77) increased the risk of AAU. The incidence rate ratio (IRR) of AAU with etanercept was significantly higher than that of adalimumab (IRR 1.78). The IRR of AAU was also higher for etanercept than adalimumab use in patients with (IRR 1.86) and without (IRR 2.92) a history of AAU. Conclusion: These data suggest that compared to anti-TNF-alpha monoclonal antibodies, etanercept has a higher incidence of AAU regardless of a history of AAU.

## 1. Introduction

Ankylosing spondylitis (AS) is a chronic inflammatory disease that mainly affects the spine, sacroiliac joints, and peripheral joints and it is often accompanied by extra-articular manifestations [1]. Acute anterior uveitis (AAU) is the most common extra-articular manifestation in AS, with a pooled prevalence of 25.8% according to a recent meta-analysis [2]. AAU tends to recur and can cause visual impairment and ocular complications, one of the major causes of a poor quality of life in patients with AS [3].

Tumor necrosis factor-alpha (TNF-α) is a pro-inflammatory cytokine that is critical in maintaining host defenses; it also plays a key role in the pathogenesis of several chronic inflammatory diseases, including AS [4]. The introduction of TNF-α inhibitors (TNFis) has led to a significant improvement in the treatment of AS. Several studies have reported the efficacy of TNFis in reducing the incidence rate (IR) of AAU in patients with AS [5,6]. However, among TNFis, soluble TNF receptors and anti-TNF-α monoclonal antibodies have shown different IRs of AAU [7,8,9,10]. A previous study also found that a history of AAU is more strongly associated with the occurrence of AAU than the type of TNFis [10]. Since AAU is managed by ophthalmologists, it is difficult for rheumatologists to detect AAU in the clinic. Thus, a study on the IR of AAU during the use of TNFis in patients with AS would be difficult with a hospital-based cohort.

Therefore, the present study aimed to evaluate the occurrence of AAU in patients with AS who were being treated with different types TNFis by using a nationwide claims database of the Korean Health Insurance Review and Assessment Service (HIRA). The IRs of AAU were also compared for each TNFis according to their history of AAU.

## 2. Materials and Methods

### 2.1. Study Design and Participants

This was a population-based retrospective cohort study using patient records extracted from the HIRA claims database. The database includes all of the health-related information of around 50 million people in the entire South Korean population, as covered by the National Health Insurance (NHI) program [11]. It contains information on patient demographics, diagnosis (using the International Statistical Classification of Diseases and Related Health Problems, 10th revision, [ICD-10]), medical procedures, and prescriptions. The prescription data include brand and generic drug names according to the HIRA drug formulary code, prescription date, supply date, and route of administration [12]. All of the recorded diagnoses were based on the ICD-10 and Korean rare intractable disease (RID) registration code for AS.

We identified new users of TNFis among patients diagnosed with AS in the claims database between January 2009 and December 2019. First, we identified all of the patients with AS from the Korean NHI Claims database from January 2009 to December 2019. Patients with AS (ICD-10, M45) or with the Korean RID registration code for AS (V140) were included. In the Korean RID system, AS was diagnosed according to the 1984 Modified New York criteria that comprised the radiographic evidence of sacroiliitis and clinical symptoms [13]. We excluded patients who had already been diagnosed with AS between 2007 and 2008 in order to screen for new patients. Patients diagnosed with systemic lupus erythematosus (ICD-10, M32), rheumatoid arthritis (RID code, V223), Behcet’s disease (ICD-10, M352), psoriatic arthritis (ICD-10, M07), Crohn’s disease (ICD-10, K50), and/or ulcerative colitis (ICD-10, K51) before the diagnosis of AS were excluded for diagnostic validation. We included patients who initiated TNFis (including etanercept [ETN], which is a soluble TNF receptor, and adalimumab [ADA], infliximab [IFX], and golimumab [GOL], which are anti-TNF-α monoclonal antibodies) during this period and maintained it for at least 2 years (730 days).

### 2.2. Procedures and Outcomes

The primary endpoint of this study was the occurrence of the first event of AAU in patients with AS during the first 2 years of treatment with each of the TNFis (ETN, ADA, IFX, and GOL) with or without a history of AAU. The secondary endpoints of this study were the total occurrence, including recurrence of AAU during the first 2 years of TNFis treatment. The first use of ETN, ADA, IFX, or GOL was defined as the TNFi index date. Since the information on uveitis activity could not be retrieved due to the nature of the claim-based dataset, we defined the occurrence of AAU as patients who were treated using prescriptions for cyclosporin or steroids (oral/injection/topical) during the same visit that they were diagnosed with a diagnostic code of AAU (ICD-10 codes H200, H201, H208, H209, and H221). According to the Standardization of Uveitis Nomenclature Working Group, the term “recurrent uveitis” describes episodes of uveitis separated by at least 3 months of inactivity without treatment [14]. In this study, we aimed to identify claims of AAU with treatment (steroids or cyclosporin) separated by at least 3 months.

The patients were classified into two groups according to the presence or absence of a history of AAU. Patients with a history of AAU within 2 years prior to the diagnosis of AS were classified as “history of AAU”.

Additional clinical data collected from these patients included age, sex, comorbidities, Charlson comorbidity index [15], and the use of medications. Previous diagnoses of the following diseases were noted: dyslipidemia, hypertension, diabetes mellitus, chronic obstructive pulmonary disease, ischemic heart disease, psoriasis, renal failure, and asthma. Recent use of the following medications was identified: steroids, non-steroidal anti-inflammatory drugs (NSAIDs), and immune-modulating agents (including sulfasalazine [SSZ], methotrexate [MTX], cyclosporine, azathioprine, mycophenolate mofetil, and cyclophosphamide). These comorbidities and medications used by the study population were investigated by searching for standardized codes in the database (Appendix A). AS-related medications of the study population were investigated by searching for HIRA formulary codes in the database (Appendix A).

### 2.3. Statistical Analysis

Characteristics of the included patients with AS, such as age, comorbidities, and medications, were analyzed. Categorical and continuous variables were presented as numbers with percentages and mean with standard deviation, respectively. The IR with a 95% confidence interval (CI) per 100 person-years of the first AAU events was estimated by applying the Poisson distribution. The hazard ratio (HR) and 95% CIs of the first AAU was calculated for each TNFis Cox regression model. The risk of AAU was also estimated with multivariable adjustment for age, sex, use of AS medications, and the presence of comorbidities, such as hypertension, stroke, chronic obstructive pulmonary disease, and diabetes mellitus, as appropriate. Kaplan-Meier curves were used to assess the cumulative AAU IR for each TNFis group, and the log-rank test was used to detect differences. To compare the overall IRs of AAU, including recurrence between TNFis users, the IRs per 100 person-years with 95% CIs in both groups were calculated and compared as the IR ratio (IRR) to fit the Poisson regression. The risk of total AAU occurrence was estimated with multivariable adjustment for the use of medications (MTX and SSZ) within 1 year of initiation of TNFis, as appropriate. Sensitivity analysis was performed by analyzing each TNFis for the first occurrence and the total occurrence (the first and recurrence) of AAU. *p* < 0.05 was considered statistically significant. All of the statistical analyses were performed using SAS Enterprise Guide software (version 6.1; SAS Institute, Inc., Cary, NC, USA) and R software (version 3.5.1; R Foundation for Statistical Computing, Vienna, Austria).

## 3. Results

### 3.1. Study Population

We identified 88,904 individuals with the AS diagnostic code in the Korean NHI claim database between 2009 and 2019. Among them, we excluded 17,540 individuals who were previously diagnosed with AS between 2007 and 2008. An additional 9998 individuals with other rheumatic diseases (systemic lupus erythematosus, rheumatoid arthritis, Behcet’s disease, psoriatic arthritis, Crohn’s disease, and ulcerative colitis) diagnosed before a diagnosis of AS were also excluded. Another 53,392 patients who had never used TNFis and 2036 patients who used TNFis for less than 2 years prior to the study were excluded. The final cohort comprised 5938 patients with newly diagnosed AS who maintained TNFis treatment for more than 2 years (Figure 1).

As shown in Table 1, the most frequently used TNFi was ADA (2477, 41.7%), followed by ETN (1218, 20.5%), IFX (1214, 20.4%), and GOL (1029, 17.3%). Among these TNFi users, 1488 (25.1%) had a history of AAU before starting treatment. The mean patient age was 37.2 years; 77.6% of the patients were male. Regular use of NSAIDs during TNFi treatment was significantly lower in the ADA group. Furthermore, the exposure to corticosteroid and immune modulatory agents was lower in the GOL group (Table 2).

### 3.2. The Risk of AAU According to the Type of TNFis

As shown in Table 3, the first occurrence of AAU within 2 years of using TNFi was found in 609 patients. The crude IR for the development of AAU in patients under TNFis treatment was 5.4 (95% CI: 5.0–5.9) per 100 person-years. The IR per 100 person-years were 4.8 (95% CI 4.2–5.5) for ADA, 8.4 (95% CI 7.3–9.7) for ETN, 4.5 (95% CI 3.7–5.5) for IFX, and 4.6 (95% CI 3.7–5.6) for GOL. In the adjusted Cox regression analysis, ETN was more strongly associated with the development of AAU than ADA (HR 1.77, 95% CI 1.46–2.14, *p* < 0.001), while there was no statistically significant difference among ADA, IFX, and GOL. Among the patients with a history of AAU who were uveitis-free during the 2 years before TNFis initiation, ETN was also associated with a higher adjusted HR than was ADA (AAU history (+): HR 1.99, 95% CI 1.58–2.50, *p* < 0.001; AAU history (−): HR 2.82, 95% CI 1.92–4.13, *p* < 0.001). 

As revealed by the Kaplan-Meier curve (Figure 2), in the patients with and without a history of AAU, the cumulative AAU IR of ETN users significantly increased compared with that of anti-TNF-α monoclonal antibodies users (log-rank test, *p* < 0.05).

### 3.3. Incidence of All AAU Events within 2 Years after the Initiation of TNF Inhibitors

Table 4 shows the IRR of total AAU occurrences within 2 years of TNFis use. The adjusted IRR for AAU in patients with ETN was 1.78 (95% CI: 1.46–2.18, *p* < 0.001) compared with that in patients with ADA. In addition, the IRR of AAU was also higher in those taking ETN than in those taking ADA in patients with and without a history of AAU (IRR 1.86, 95% CI 1.51–2.29, *p* < 0.001 and 2.92, 95% CI 2.00–4.26, *p* < 0.001). The rate of recurrence of AAU more than two times was 88 (33.0%) in ADA, 96 (42.9%) in ETN, 36 (30.3) in IFX, and 32 (31.1%) in GOL.

## 4. Discussion

This large population-based cohort study using the Korean NHI Claims database demonstrated that the AAU incidence after TNFi initiation was higher in patients with AS under ETN treatment than those treated with anti-TNF-α monoclonal antibodies, including ADA. AAU occurred in nearly 50% of patients with a history of AAU who had used ETN for 2 years, twice that of patients treated with anti-TNF-α monoclonal antibodies.

Uveitis accounts for between 10% and 15% of preventable vision loss and is the third leading cause of blindness worldwide [16]. AAU approximately occurs in between 25% and 40% of patients with AS [2,17]. Several types of TNFis are currently used to treat patients with AS, including anti-TNF-α monoclonal antibodies (ADA, IFX, GOL, and certolizumab) and the soluble TNF receptor molecule (ETN) [18]. Many studies have reported the efficacy of TNFis in reducing the incidence of AAU in patients with AS. However, some studies have suggested that TNFis may actually induce new-onset AAU [19,20]. Although inflammatory arthritis improves after TNFis use, AAU can occur regardless of AS disease activity [20]. The results of several studies are still inconclusive as to which and how TNFis influence the development of AAU. Although some studies have shown that anti-TNF-α monoclonal antibodies and ETN similarly prevent AAU [5,21], many studies have reported that, compared to ETN, anti-TNF-α monoclonal antibodies are better at preventing AAU [7,9,22,23]. For ETN, the possibility of new-onset uveitis following TNFis therapy has been reported, usually in patients treated with ETN [20,24,25]. However, compared to ETN, IFX was found to increase the risk of AAU to a greater extent in a Korean single center study [26].

Although the precise effect of TNFi on AAU in patients with AS still cannot be concluded, the prevention of AAU in patients with AS is expected to be more effective with anti-TNF-α monoclonal antibodies than with ETN.

The mechanisms underlying whether ETN is associated with more cases of AAU than anti-TNF-α monoclonal antibodies are still unclear. There are several hypotheses about the differences in the occurrence of uveitis between those taking anti-TNF-α monoclonal antibodies and those taking soluble TNF receptor (ETN). In terms of pharmacokinetics, ETN has a shorter half-life, a shorter volume of distribution, and a faster clearance than ADA and IFX [27]. TNF-α has both a membrane-bound form and a soluble form. TNF receptor 1 (TNFR1, also known as p55), which is present on most cell types, is the main receptor for both the membrane bound and soluble form of TNF-α. By comparison, TNFR2 (p75) is mainly present on cells of the immune system (including retinal macrophages) and only responds to membrane bound TNF-α [28]. The anti-TNF-α monoclonal antibodies ADA, IFX, and GOL can bind to both TNF receptors (TNFR1 and TNFR2), whereas ETN mainly blocks soluble TNF-α and has low affinities to membrane-bound TNF-α [29]. Distinct from anti-TNF-α monoclonal antibodies, ETN also has the capacity to efficiently bind and then release TNF-α, which may serve to prolong the circulating half-life of TNF-α, leading to prolonged intraocular TNF-α stimulating uveitis [28].

Results from the Swedish biologics register reported that ETN is significantly associated with a higher HR of uveitis (3.86) compared to ADA and 1.99 compared to IFX [9]. Another observational study using a US database of patients with spondyloarthritis reported a risk of uveitis 1.9-fold higher in the first year with ETN treatment as compared with ADA treatment [7]. Similarly, in the present study, the frequency of a first occurrence of AAU within 2 years after the initiation of TNFis was significantly higher in the ETN group (HR 1.81) than in the anti-TNF-α monoclonal antibodies (ADA, IFX, and GOL) group, and IRR, including the recurrence of AAU within 2 years, was also 1.85-fold higher in the ETN group than in the anti-TNF-α monoclonal antibodies group.

On the other hand, a recent observational study reported that the risk of uveitis in patients with spondyloarthritis was higher with the use of ETN than of anti-TNF-α monoclonal antibodies, but the differences were not significant. Additionally, it was suggested that, compared to the type of TNFis, a history of uveitis was more strongly related to the occurrence of uveitis [10]. In the present study, the risk of developing AAU during ETN treatment was approximately twice that of ADA use in patients with a history of AAU, despite it being three times higher in patients without a history of AAU. In particular, the Kaplan Meier curve (Figure 2) showed that in patients with a history of AAU, AAU recurred in nearly 50% of patients under ETN treatment within 2 years of its initiation. In this study, the HR and IRR of AAU when using ETN were relatively lower in the groups with a previous history of AAU compared to those without a history of AAU. We speculated that anti-TNF-α monoclonal antibodies may have been preferred by the prescribing physicians over ETN in the group of patients with a recurrent history of AAU when selecting TNFis.

The present study had some limitations. First, the NHI claims database lacks detailed clinical information on individual patients, such as laboratory findings, disease activities, radiological findings, and lifestyle factors. Second, the AAU occurrences were identified based on the registered diagnoses code, which could not reflect the severity of AAU. Third, certolizumab pegol was excluded from the analysis because it is not available in Korea. Fourth, in Korea, TNFis preparations can be prescribed as insurance benefits for patients with sacroiliitis according to the 1984 Modified New York criteria for patients only to those with active AS who show an insufficient therapeutic effect even after receiving two or more NSAIDs for more than 3 months. Therefore, there is a limitation in that early AS patients or non-radiographic axial spondyloarthritis patients are not included. Nevertheless, the major strength of this study was that it utilized the largest possible population using a nationwide claims database to report the AAU incidence in AS patients after TNFis initiation. We also explored the incidence of AAU by types of TNFis and history of AAU.

## 5. Conclusions

This nationwide cohort study will provide a better understanding of the association between TNFis and AAU. The incidence of AAU was higher in AS patients treated with ETN than in cases treated with anti-TNF-α monoclonal antibodies, including ADA, IFX, and GOL. Based on our results, it could be emphasized that anti-TNF-α monoclonal antibodies are more appropriate for AS patients with a history of AAU to prevent a recurrent event.

## Figures and Tables

**Figure 1 jcm-11-00631-f001:**
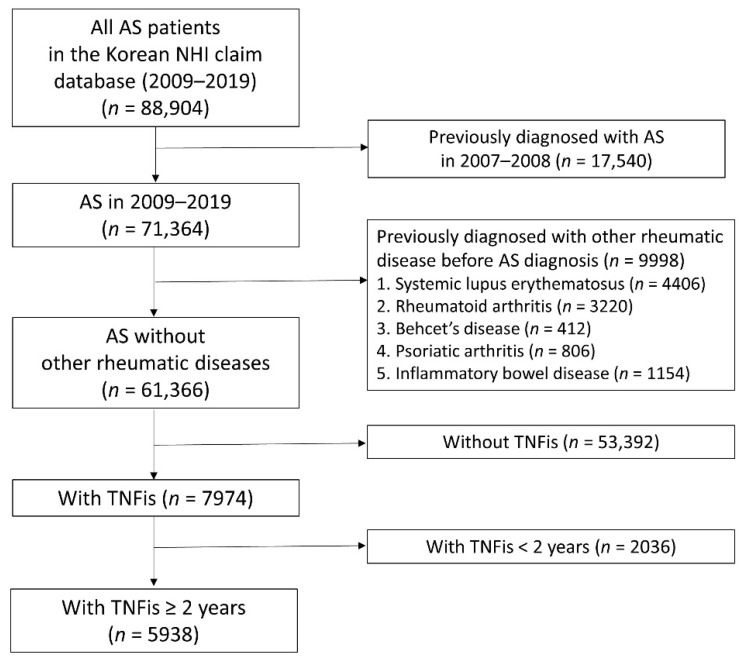
Selection of the study population. AS: ankylosing spondylitis, NHI: National Health Insurance, TNFis: tumor necrosis factor-alpha inhibitors.

**Figure 2 jcm-11-00631-f002:**
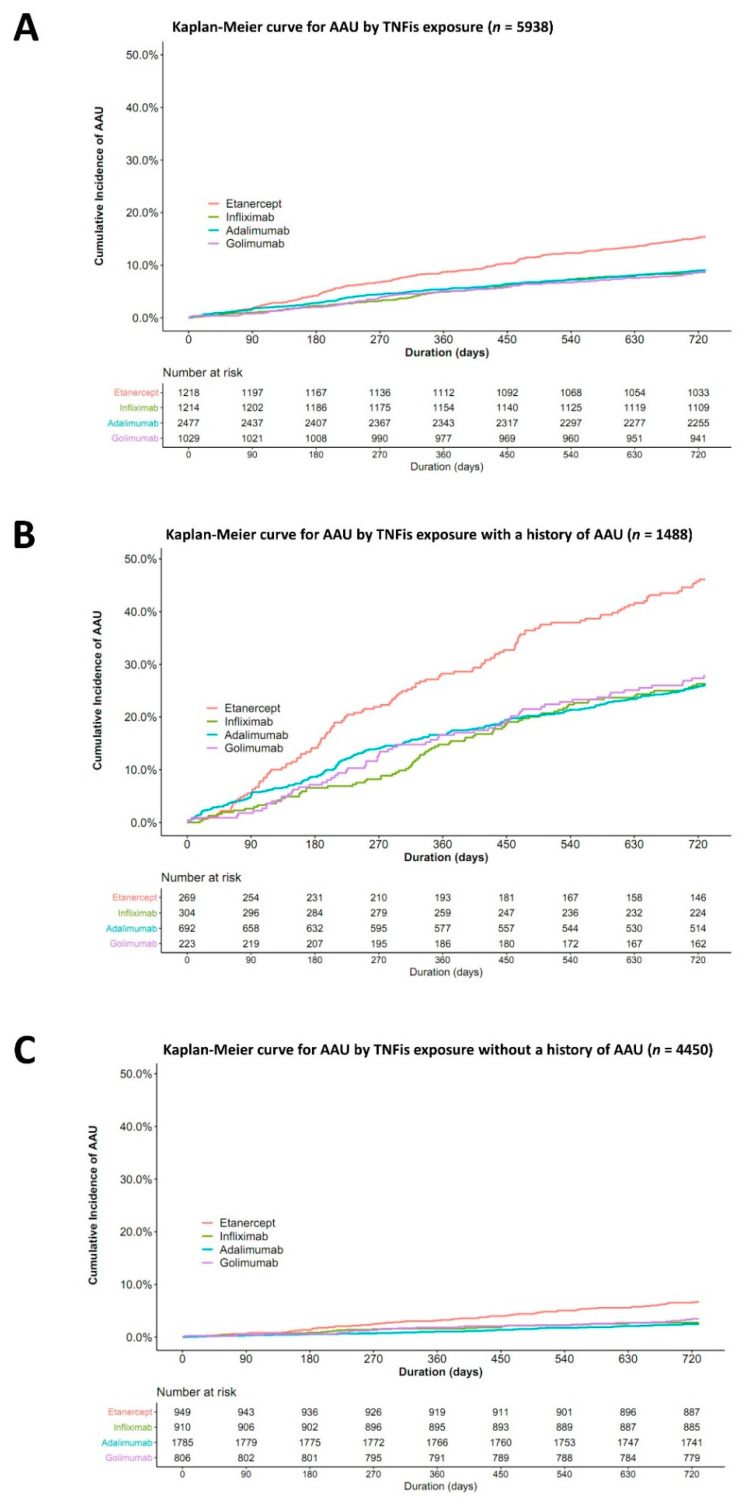
Cumulative probability of the occurrence of acute anterior uveitis after tumor necrosis factor alpha inhibitor initiation in overall patients (**A**) and in patients with (**B**) or without (**C**) a history of acute anterior uveitis. AAU: acute anterior uveitis, TNFis: tumor necrosis factor-alpha inhibitors.

**Table 1 jcm-11-00631-t001:** Baseline characteristics of TNF-α inhibitor users.

	Total (*n* = 5938)	ADA (*n* = 2477)	ETN (*n* = 1218)	IFX (*n* = 1214)	GOL (*n* = 1029)	*p*-Value
Treatment duration, year	4.7 ± 2.2	4.8 ± 2.3	4.8 ± 2.3	5.3 ± 2.3	3.6 ± 1.3	<0.001
Age, year	37.2 ± 13.1	36.8 ± 13.2	37.6 ± 14.8	37.0 ± 13.1	38.0 ± 13.6	0.083
Age group, year						0.143
<30	1983 (33.4)	839 (33.9)	409 (33.6)	405 (33.4)	330 (32.1)	
30–39	1537 (25.9)	654 (26.4)	286 (23.5)	325 (26.8)	272 (26.4)	
40–49	1239 (20.9)	522 (21.1)	254 (20.9)	251 (20.7)	212 (20.6)	
50–59	793 (13.4)	326 (13.2)	170 (14.0)	161 (13.3)	136 (13.2)	
60<	386 (6.5)	136 (5.5)	99 (8.2)	72 (5.9)	79 (7.6)	
Sex						0.041
Male	4610 (77.6)	1959 (79.1)	953 (78.2)	915 (75.4)	783 (76.1)	
Female	1328 (22.4)	518 (20.9)	265 (21.8)	299 (24.6)	246 (23.9)	
Previous AAU history	1488 (25.1)	692 (27.9)	269 (22.1)	304 (25.0)	223 (21.7)	<0.001
Charlson comorbidity index	0.53 ± 0.97	0.52 ± 0.93	0.61 ± 1.11	0.47 ± 0.84	0.57 ± 1.04	0.002
0	3939 (66.3)	1648 (66.5)	786 (64.5)	825 (68.0)	680 (66.1)	0.001
1 to 2	1734 (29.2)	730 (29.5)	363 (29.8)	356 (29.3)	285 (27.7)	
3 to 4	265 (4.5)	99 (4.0)	69 (5.7)	33 (2.7)	64 (6.2)	
Comorbidities (1 year prior to index date)						
Dyslipidemia	756 (12.7)	307 (12.4)	170 (14.0)	154 (12.7)	125 (12.1)	0.528
Hypertension	702 (11.8)	298 (12.0)	145 (11.9)	132 (10.9)	127 (12.3)	0.699
Diabetes mellitus	305 (5.1)	127 (5.1)	72 (5.9)	53 (4.4)	53 (5.2)	0.394
COPD	226 (3.8)	94 (3.8)	55 (4.5)	41 (3.4)	36 (3.5)	0.466
Ischemic heart disease	118 (2.0)	54 (2.2)	27 (2.2)	18 (1.5)	19 (1.8)	0.477
Psoriasis	61 (1.0)	26 (1.0)	9 (0.7)	14 (1.2)	12 (1.2)	0.707
Stroke	47 (0.8)	19 (0.8)	14 (1.1)	6 (0.5)	8 (0.8)	0.337
Renal failure	25 (0.4)	7 (0.3)	10 (0.8)	1 (0.1)	7 (0.7)	0.013
Asthma	21 (0.4)	10 (0.4)	7 (0.6)	0 (0.0)	4 (0.4)	0.102

Values are presented as number (%) or mean ± standard deviation. TNF-α: tumor necrosis factor-alpha, ADA: adalimumab, ETN: etanercept, IFX: infliximab, GOL: golimumab, AAU: acute anterior uveitis, COPD: chronic obstructive pulmonary disease.

**Table 2 jcm-11-00631-t002:** Use of concomitant medications with TNF-α inhibitor use.

	Total (*n* = 5938)	ADA (*n* = 2477)	ETN (*n* = 1218)	IFX (*n* = 1214)	GOL (*n* = 1029)	*p*-Value
Use of corticosteroid (≥90 days)	2993 (50.4)	1254 (50.6)	611 (50.2)	668 (55.0)	460 (44.7)	<0.001
Use of NSAIDs (≥0.7 PDC)	2424 (40.8)	939 (37.9)	533 (43.8)	535 (44.1)	417 (40.5)	<0.001
Immune modulating agents	5117 (86.2)	2214 (89.4)	1016 (83.4)	1082 (89.1)	805 (78.2)	<0.001
Sulfasalazine	4792 (80.7)	2082 (84.1)	934 (76.7)	1011 (83.3)	765 (74.3)	<0.001
Methotrexate	1998 (33.6)	872 (35.2)	422 (34.6)	449 (37.0)	255 (24.8)	<0.001
Cyclosporine	192 (3.2)	83 (3.4)	34 (2.8)	45 (3.7)	30 (2.9)	0.558
Azathioprine	84 (1.4)	26 (1.0)	16 (1.3)	35 (2.9)	7 (0.7)	<0.001
Mycophenolate mofetil	12 (0.2)	6 (0.2)	1 (0.1)	5 (0.4)	0 (0.0)	0.122
Cyclophosphamide	2 (0.0)	0 (0.0)	0 (0.0)	2 (0.2)	0 (0.0)	0.051

Values are presented as number (%) or mean ± standard deviation. TNF-α: tumor necrosis factor-alpha, ADA: adalimumab, ETN: etanercept, IFX: infliximab, GOL: golimumab, NSAIDs: non-steroidal anti-inflammatory drugs; PDC: proportion of days covered.

**Table 3 jcm-11-00631-t003:** The risk of the first AAU occurrence within 2 years of initiation of each TNF-α inhibitor.

	Total	AAU Cases	Sum of py	IR (/100 py)	95% CI	HR	95% CI	*p*-Value	aHR *	95% CI	*p*-Value
**Total**	5938	609	11203	5.4	5.0–5.9						
Adalimumab	2477	226	4696	4.8	4.2–5.5	Ref			Ref		
Etanercept	1218	188	2232	8.4	7.3–9.7	1.75	1.44–2.12	<0.001	1.77	1.46–2.14	<0.001
Infliximab	1214	105	2312	4.5	3.7–5.5	0.94	0.75–1.19	0.620	0.92	0.73–1.16	0.495
Golimumab	1029	90	1962	4.6	3.7–5.6	0.95	0.75–1.22	0.698	0.98	0.76–1.25	0.846
**AAU history (+)**	1488	448	2469	18.1	16.5–19.9						
Adalimumab	692	182	1170	15.6	13.4–18.0	Ref			Ref		
Etanercept	269	124	398	31.2	25.9–37.2	1.99	1.58–2.50	<0.001	1.99	1.58–2.50	<0.001
Infliximab	304	80	523	15.3	12.1–19.0	0.98	0.75–1.27	0.876	0.97	0.74–1.26	0.807
Golimumab	223	62	379	16.4	12.6–21.0	1.05	0.79–1.40	0.739	1.07	0.80–1.43	0.647
**AAU history (−)**	4450	161	8733	1.8	1.6–2.2						
Adalimumab	1785	44	3526	1.2	0.9–1.7	Ref			Ref		
Etanercept	949	64	1834	3.5	2.7–4.5	2.80	1.91–4.11	<0.001	2.82	1.92–4.13	<0.001
Infliximab	910	25	1789	1.4	0.9–2.1	1.12	0.69–1.83	0.651	1.10	0.67–1.79	0.709
Golimumab	806	28	1584	1.8	1.2–2.6	1.42	0.88–2.28	0.150	1.45	0.90–2.32	0.128

AAU: acute anterior uveitis, TNF-α: tumor necrosis factor-alpha, py: person-years, IR: incidence rate, CI: confidence interval, HR: hazard ratio, aHR: adjusted hazard ratio, Ref: reference. * Adjusted for age, Charlson comorbidity index, use of steroids, presence for hypertension and stroke in total, and adjusted age, use of steroid in AAU history (+), and adjusted for the use of steroids, the presence of chronic obstructive pulmonary disease, and diabetes mellitus in the AAU history (−).

**Table 4 jcm-11-00631-t004:** The risk of occurrence of total AAU events within 2 years of initiation of each TNF-α inhibitor.

	AAU Cases	Sum of py	IR (/100 py)	95% CI	IRR	95% CI	*p*-Value	IRR *	95% CI	*p*-Value
**Total**										
Adalimumab	389	5612	6.8	5.9–7.8	Ref			Ref		
Etanercept	355	2760	12.4	10.7–14.3	1.82	1.49–2.22	<0.001	1.78	1.46–2.18	<0.001
Infliximab	168	2620	6.2	5.0–7.6	0.91	0.70–1.17	0.456	0.92	0.72–1.18	0.525
Golimumab	145	2232	6.5	5.3–8.0	0.96	0.75–1.22	0.712	0.92	0.72–1.18	0.512
**AAU history (+)**										
Adalimumab	313	1549	19.9	17.2–22.9	Ref			Ref		
Etanercept	230	602	37.6	32.5–43.6	1.89	1.54–2.33	<0.001	1.86	1.51–2.29	<0.001
Infliximab	132	652	19.8	16.0–24.5	1.00	0.77–1.29	0.985	1.00	0.78–1.30	0.986
Golimumab	106	488	21.4	17.2–26.8	1.08	0.83–1.40	0.568	1.05	0.81–1.36	0.702
**AAU history (−)**										
Adalimumab	76	4063	1.9	1.4–2.5	Ref			Ref		
Etanercept	125	2159	5.5	4.3–7.1	2.98	2.04–4.37	<0.001	2.92	2.00–4.26	<0.001
Infliximab	36	1969	1.8	1.2–2.6	0.95	0.57–1.58	0.852	0.97	0.59–1.60	0.920
Golimumab	39	1745	2.3	1.6–3.3	1.22	0.76–1.95	0.417	1.16	0.72–1.87	0.554

AAU: acute anterior uveitis, TNF-α: tumor necrosis factor-alpha, py: person-years, IR: incidence rate, IRR: incidence rate ratio, CI: confidence interval, HR: hazard ratio, aHR: adjusted hazard ratio, Ref: reference. * Adjusted for the use of disease-modifying anti-rheumatic drugs (methotrexate, sulfasalazine).

## Data Availability

The datasets generated and/or analyzed during the current study are not publicly available due to Data Protection Laws and Regulations in Korea, but the final analyzed results are available from the corresponding authors upon reasonable request.

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
