# Peer review of "Risk of Acute Anterior Uveitis in Ankylosing Spondylitis According to the Type of Tumor Necrosis Factor-Alpha Inhibitor and History of Uveitis: A Nationwide Population-Based Study"

_jcm, 2022, doi:10.3390/jcm11030631_

Round 1
Reviewer 1 Report
Comments to the Authors jcm-1568127
Title: Risk of acute anterior uveitis in ankylosing spondylitis according to the type of tumor necrosis factor-alpha inhibitor and history of uveitis: a nationwide population-based study.
The authors presented results of the nationwide cohort study evaluating the risk of acute anterior uveitis (AAU) in patients with ankylosing spondylitis (AS) during treatment with tumor necrosis factor-alpha inhibitors (TNFis).
The limitation of the study is that all the information is retrospective, based on the NIH database, and no clinical or laboratory data is available.
The topic is not new, but interesting and important. The presented research is comprehensive and precise.
I have minor comments to Authors:
- I do suggest to change the text of Abstract, similar to Conslusions: “The incidence of AAU was higher in AS patients treated with ETN than in cases with anti-TNF-α monoclonal antibodies, including ADA, IFX, and GOL”.
- The English language of the manuscript needs verification. I do suggest correction of the manuscript by a native speaker.
Author Response
Please see the attachment.
Thank you for reviewing our study in meticulous detail and providing helpful comments.

Reviewer 2 Report
AAU often appears before AS diagnostic is performed; that means in early onset. Why patients with MRI sacroiliitis were not included?
What initiating time for biologic therapy was accepted?
TNFI original or biosimilars? Was there a difference made?
“Among the patients with a history of AAU and who were uveitis-free during 165 the 2 years before TNFis initiation, ETN was also associated with a higher adjusted HR 166 than was ADA (AAU history (+): HR 1.99, 95% CI 1.58–2.50, p < 0.001; AAU history (-): 167 HR 2.82, 95% CI 1.92–4.13, p < 0.001).” Does it mean during the 2 years of initiation of TNF Therapy?
It would be interesting to know if patients with AAU occurrence had a higher disease activity? Because of the use of comedication, especially corticosteroids, it looks like.
Author Response

(The authors gave the same response as above.)
